# Effectiveness of Nutritional Ingredients on Upper Gastrointestinal Conditions and Symptoms: A Narrative Review

**DOI:** 10.3390/nu14030672

**Published:** 2022-02-05

**Authors:** Rebekah M. Schulz, Nitin K. Ahuja, Joanne L. Slavin

**Affiliations:** 1Department of Food Science and Nutrition, University of Minnesota, St. Paul, MN 55108, USA; schu4398@umn.edu; 2Division of Gastroenterology and Hepatology, University of Pennsylvania, Philadelphia, PA 19104, USA; nitin.ahuja@pennmedicine.upenn.edu

**Keywords:** fiber, botanicals, herbs, dyspepsia, upper gastrointestinal, GERD, heartburn, reflux, nausea

## Abstract

Nutritional ingredients, including various fibers, herbs, and botanicals, have been historically used for various ailments. Their enduring appeal is predicated on the desire both for more natural approaches to health and to mitigate potential side effects of more mainstream treatments. Their use in individuals experiencing upper gastrointestinal (GI) complaints is of particular interest in the scientific space as well as the consumer market but requires review to better understand their potential effectiveness. The aim of this paper is to review the published scientific literature on nutritional ingredients for the management of upper GI complaints. We selected nutritional ingredients on the basis of mentions within the published literature and familiarity with recurrent components of consumer products currently marketed. A predefined literature search was conducted in Embase, Medline, Derwent drug file, ToXfile, and PubMed databases with specific nutritional ingredients and search terms related to upper GI health along with a manual search for each ingredient. Of our literature search, 16 human clinical studies including nine ingredients met our inclusion criteria and were assessed in this review. Products of interest within these studies subsumed the categories of botanicals, including fiber and combinations, and non-botanical extracts. Although there are a few ingredients with robust scientific evidence, such as ginger and a combination of peppermint and caraway oil, there are others, such as melatonin and marine alginate, with moderate evidence, and still others with limited scientific substantiation, such as galactomannan, fenugreek, and zinc-l-carnosine. Importantly, the paucity of high-quality data for the majority of the ingredients analyzed herein suggests ample opportunity for further study. In particular, trials with appropriate controls examining dose–response using standardized extracts and testing for specific benefits would yield precise and effective data to aid those with upper GI symptoms and conditions.

## 1. Introduction

### 1.1. Background on Upper GI Issues

American adults have been experiencing an increasing prevalence of upper gastrointestinal (GI) symptoms, whether in an occasional or chronic manner [1]. A global survey indicates that 40% of adults worldwide suffer with one of many functional gastrointestinal disorders (FGIDs), with upper GI conditions representing a large component of those disorders [2]. A Cedars-Sinai study showed that two of every five Americans suffer from gastro-esophageal reflux disease (GERD)-like symptoms every week, including heartburn and regurgitation [3,4].

Examples of structural upper GI conditions include erosive esophagitis, gastritis, and peptic ulcer disease, sometimes associated with *Helicobacter pylori* (*H. pylori*) infection [5,6]. Functional disorders involving the upper GI tract include GERD, esophageal hypersensitivity, esophageal dysmotility, and non-ulcer dyspepsia. FGIDs, now more commonly categorized as disorders of gut–brain interaction (DGBIs), are clinically defined per the Rome IV criteria [7]. Particularly common DGBIs of the upper GI tract include functional dyspepsia (FD), marked by symptoms of epigastric abdominal pain with or without early satiety, nausea or vomiting; and non-cardiac chest pain (NCCP). Disorders of the lower GI tract will not be included in this review.

Among these upper GI conditions, symptoms can include heartburn, reflux, indigestion, regurgitation, nausea, vomiting, trouble swallowing, upper abdominal fullness, bloating, loss of appetite, and epigastric pain. While some individuals may experience occasional symptoms of upper GI distress, which may differ only slightly from normal perceived baselines, other individuals may notice a consistent pattern of these symptoms, which can then be mapped to a defined DGBI diagnosis [7].

Addressing upper GI conditions and symptoms is important due not only to their high healthcare utilization costs (annual expenditure of $135.9 billion in the United States), but also their impact on health-related quality of life (QoL), hindrance of ability to work, and necessity for prescriptions and/or over-the-counter (OTC) medications [2,3,8]. While current treatments for functional GI conditions and symptoms include H2 receptor antagonists (H2RA), proton pump inhibitors (PPIs), antidepressants, and prokinetics, there is interest from consumers in more natural-based products, which is accompanied to some extent by published evidence in the scientific literature [3].

### 1.2. Background on Nutritional Ingredients and Use for Upper GI Support

Nutritional ingredients (a collective term used herein to describe botanicals, including dietary fibers, and non-botanicals) have been employed throughout history for a myriad of GI conditions and symptoms due to their claimed anti-ulcer, carminative, spasmolytic, soothing, and laxative effects, to name a few [9]. The usage of these products continues to increase, as evidenced by sales growth of 8.6% in 2019, the second largest percentage increase since 1998 [10,11].

To assess the state of the field and identify potentially fruitful avenues of future investigation, we chose to examine the clinical evidence for key ingredients’ effectiveness in treating upper GI conditions and symptoms. Table 1 provides a list of these 25 nutritional ingredients.

### 1.3. Primary Aim of This Review

The primary aim of this review is to assess the robustness of clinical evidence and other data on the use of a select number of popular nutritional ingredients for upper GI conditions and symptoms.

## 2. Materials and Methods

This narrative review was conducted following principles from the Preferred Reporting Items for Systematic Reviews and Meta-Analyses (PRISMA) statement [12].

In developing the research question, the PICOS (population, intervention, comparator, outcome, and study design) criteria were used as well as consultation amongst authors to establish research design, study eligibility, and inclusion/exclusion criteria (Table 2) [13]. The 25 ingredients were identified as being of interest based on currently marketed ingredients and information in the scientific literature.

To better understand how these nutritional ingredients may impact upper GI health, a literature search with predefined search criteria was conducted in June 2021 using the Embase, Medline, Derwent drug file, Pubmed, and ToXfile databases. MeSH (Medical Subject Headings) terms and keywords for nutritional ingredients as well as common upper GI complaints and symptoms were used in the search string (Table 3). The literature search was conducted with a time limit of 15 years (2006).

For the initial search, the librarian screened for duplicates and then titles and abstracts to determine suitability for inclusion. They included all suitable titles and abstracts in a document that the primary author (RS) further screened for suitability. A second reviewer (RK) checked off on this process. Hand-searched articles were also checked off by a second reviewer. Full-text articles were screened independently by one review author and independently checked by a second reviewer.

The reviewers used a standard data collection form for study characteristics and outcome data. One review author (RS) extracted the following study characteristics from the included studies: Methods: study design, total duration of study, and number and location of study centers; Participants: number of participants, mean age, age range, gender, severity of condition, and diagnostic criteria; Interventions: intervention, comparison, duration of treatment, and follow-up time; Outcomes: primary and secondary outcomes specified and collected. Any discrepancies were resolved through discussion between RK and RS.

## 3. Results and Discussion

### 3.1. Identified Trials

A total of 2282 articles were identified in the initial search strategy of 25 ingredients. After duplicates were removed, 1958 articles remained. The initial screening removed 1746 articles and the remaining abstracts that were included for further screening by the primary author totaled 212. This literature search process is outlined in Figure 1.

The primary author then further screened the full text of the 212 abstracts and titles and excluded 200 for the following reasons: the paper was not related to upper GI symptoms (39); it examined an irrelevant intervention (20); its language of publication was not English (5); it examined animals (1); it examined child/adolescent subjects (14); it was a secondary analysis (1); it was a duplicate (3), the full text was unavailable (3); it was a review (61); it was observational (2); or because it had been listed in a recent review (51). From this search, 12 articles were eligible and included in this review.

The manual search method yielded four more eligible articles which were included and assessed by the second reviewer. This process resulted in 16 articles covering nine ingredients that were eligible and included in this review. The additional 16 ingredients either lacked sufficient evidence to be mentioned, defined by possessing only animal or in vitro trials, or are reviewed in the discussion as ingredients meriting future investigation. The included ingredients fell into two main categories: botanical (including fiber, “other” botanicals, and combinations) and non-botanical.

### 3.2. Botanical Ingredients Addressing Heartburn, GERD, and Gastric Conditions: Fiber, Other Botanicals, and Combinations

#### 3.2.1. Fiber

Fiber has been extensively studied for its impacts on lower GI health and recently more efforts have been invested in exploring fiber for its upper GI benefits, including two ingredients (fenugreek and galactomannan, reviewed herein and included in the table), and a third, marine alginate, which we mention only briefly due to its having recently been extensively reviewed (Table 4) [14,15].

##### Fenugreek and Galactomannan

Fenugreek is composed of a wide variety of bioactive compounds of which fiber, primarily the water-soluble fiber galactomannan, is suggested to be the effective component in observed reductions of heartburn and GERD symptoms [16]. DiSilvestro found that the fenugreek fiber group (2000 mg, twice per day, standardized to contain 85% galactomannan), and the ranitidine group (75 mg, twice per day) both yielded reduced heartburn severity and incidence in subjects (n = 45) over two weeks, and that fenugreek was significantly more effective than a placebo (Table 4) [14]. However, it is challenging to draw conclusions as to the efficacy of fenugreek based on this trial, as this was a partially blinded study and the primary outcome was measured subjectively based on participant diary entries of heartburn without pH measures, placebo, or other objective measures.

Abenavoli et al. conducted a double-blind RCT in which a formula (10 mL, three times per day) containing fenugreek-derived galactomannan significantly reduced symptoms in GERD patients (n = 60) as compared to a placebo (Table 4) [15]. However, as the formula also contained calcium carbonate and sodium bicarbonate, both known to reduce GERD symptomology, as well as *Malva sylvestris* and hyaluronic acid, indicated for mucosal healing due to their anti-inflammatory properties, it is challenging to elucidate whether and to what degree galactomannan itself is responsible for the observed effect [17,18,19,20]. Additionally, as galactomannan contains different ratios of mannose to galactose, depending on its source, further investigation is needed to determine a standardized composition [14].

The proposed mechanism for galactomannan’s effect on heartburn and GERD involves the soluble fiber forming a raft when hydrated, acting as a barrier to ameliorate the rise of acid into the esophagus, thus serving as an effective adjunct in the relief of GERD symptoms [15,21]. However, this mechanism is hypothesized based on similarities to observed MoAs for other naturally occurring polysaccharides such a marine alginate; research is needed to confirm this similarity [22]. An animal study by Pandian et al. attributed galactomannan’s superior antiulcerogenic ability to its observed reduction in gastric acid output but did not indicate a barrier mechanism [23]. Based on this evidence, fenugreek and its constituent, galactomannan, hold promise for the management of upper GI symptoms or conditions, but further investigations into a standardized extract composition and MoA are needed.

As galactomannan is a component of other gums such as guar gum, locust bean gum, and partially hydrolyzed guar gum, these could be investigated for similar effects on heartburn reduction [22,24]. In addition to galactomannan, soluble fiber as a whole is an evolving area of interest for upper GI symptom management. A prospective, open-label study (n = 30) observed an inverse relationship between increased fiber intake, specifically psyllium fiber (15 g per day), and occasional heartburn, esophageal sphincter resting pressure, and heartburn frequency in non-erosive GERD patients with previous low dietary fiber intake, defined as less than 20 g per day [25]. This trial lacked a placebo and was short in duration (10 days).

##### Marine Alginate

Marine alginate is an extract of seaweed, which, as described above, is suggested to mitigate GERD symptoms due to its raft-forming mucilage properties [21,22]. A recent SR by Zhao et al. assessed 10 RCTs, comparing various proprietary blends of marine alginate to a placebo or antacids, in combination with PPI versus PPI alone, or against PPI, in terms of their effectiveness in mitigating GERD symptoms [26]. Of these ten RCTs, seven were included in an MA. We found the authors’ use of the Cochrane Collaboration risk assessment to be satisfactory for rating the quality of evidence. Upon descriptive analysis of the related articles, they observed superior outcomes for alginate over a placebo or antacids, but pooled analysis revealed the differences in change in heartburn, regurgitation, and dyspepsia scores to be non-significant. Similarly, no significant difference was seen between changes in heartburn, regurgitation, and dyspepsia scores in a pooled analysis of those articles examining alginate in combination with PPI versus PPI alone. The articles examining alginate versus PPI were heterogenous and therefore not included in the MA, and descriptive analysis found no difference in outcomes between groups. However, although this SR grouped trials by intervention, the alginate formulations were heterogenous, and the duration of treatment varied considerably. Additionally, trials examining alginate versus a placebo or antacid should be assessed separately.

#### 3.2.2. “Other” Botanicals

Nine studies explored four “other” botanical ingredients (defined as non-fiber ingredients for the purpose of this review) and their impact on upper GI conditions: Aloe vera (A. vera), ginger, licorice, and papaya [27,28,29,30,31,32,33,34,35]. (Table 5). Here, we also review those “other” botanical ingredients with limited evidence (apple cider vinegar (ACV), cardamom, and fennel) or those subject to trials outside of the inclusion-date range (d-limonene) and thus not included in the tables.

##### Aloe Vera

A. vera is believed to help alleviate ulcerogenic and GERD related symptoms [36,37]. A 2015 open-label RCT by Panahi et al. found that A. vera syrup (10 mL once per day, standardized to 5.0 mg polysaccharide/mL syrup), when compared to ranitidine tablets (150 mg twice per day) and omeprazole capsules (20 mg once a day), was equally effective in reducing GERD symptoms in subjects (n = 70) (Table 5) [28]. A second randomized, open-label trial conducted in 2016 also by Panahi et al., found that A. vera syrup as an adjunct to pantoprazole therapy (5 mL twice per day and 40 mg once per day, respectively) was significantly more effective in reducing GERD symptoms compared to pantoprazole (40 mg once per day) alone in Iranian male veterans (n = 85) suffering from past sulfur-mustard gas exposure [27]. Although both utilized the same dosage of A. vera syrup, it is difficult to know whether both studies used a consistent composition and thus whether a dosage recommendation can be made. Furthermore, as the studies lacked placebos and were not blinded, it is difficult to make comprehensive conclusions as to A. vera’s effectiveness for GERD. Both the 2015 and 2016 studies shared adverse event (AE) data, with the earlier reporting two AEs in the A. vera group (vertigo and stomachache), neither of which resulted in dropout, and the latter reporting none.

Some point to A. vera’s antioxidant activity as its MoA related to GERD, as oxidative stress and inflammation may be involved in GERD pathogenesis [38]. In vitro and animal studies have observed reduced lipid peroxidation, hepatic and fatty markers, and necrosis upon A. vera administration [28,39,40].

##### Apple Cider Vinegar

Several studies indicate apple cider vinegar (ACV)’s usefulness in lowering postprandial glycemic response, specifically by slowing of gastric motility, which would seem paradoxical with respect to its potential value for upper GI symptoms [41,42,43]. However, these studies were small and yielded heterogenous outcomes. ACV was used in the gum formulation mentioned above, and while subjects found beneficial effects in relief of heartburn and acid reflux, it is challenging to elucidate whether this effect was due to the ACV or the other ingredients [44].

##### Cardamom

A review examining the effects of natural ingredients on PINV mentioned an RCT examining the effects of cardamom, but the trial was in Arabic and therefore unevaluable [45,46]. Two animal studies observed cardamom extract as an effective treatment for induced gastric ulcers in rats, with the proposed MoA including direct protection of the mucosal barrier as well as the extract’s ability to facilitate smooth muscle relaxation [47,48].

##### D-limonene

D-limonene, a terpene extracted from citrus essential oil *(Rutaceae)*, has a number of animal and in vitro trials highlighting it as a gastroprotective and GERD ameliorating agent [49]. Additionally, two separate clinical studies performed under a U.S. patent found that adults suffering from chronic heartburn and GERD significantly benefited from d-limonene supplementation [50]. However, we did not include this in the tables as the date was outside of our inclusion criteria. Additionally, the first trial did not contain a placebo and did not account for three of the participants, and both utilized heterogenous dosing patterns and duration of treatment, as well as differently standardized extracts (1000 mg d-limonene of 98.5–99.3 vs 98% purity, respectively).

##### Fennel

Rodent studies have shown that anethole, a chemical derivative of fennel oil, increases gastric emptying in mice, suggesting that it could be used for FD [51]. A clinical trial examined the effects of a formulation comprised of fennel, lemon balm, and German chamomile on gastric transit time, but this trial was unevaluable as infants were the studied population and thus the study fell outside of our inclusion criteria [52]. As a result, further investigation is needed to elucidate the relationship between fennel and gastric emptying in adults, and its potential impact on related upper GI conditions.

##### Ginger

Ginger’s proposed MoA is its inhibition of intestinal cholinergic M3 and serotonergic 5-HT3 receptors, manifesting in decreased nausea and vomiting, as well as increased gastric motility and a corresponding decrease in transit time [29,53,54]. Gingerols are commonly indicated as the active ingredient responsible for targeting and modulating these signaling pathways [55].

Due to the recent publication of two systematic reviews (SRs) of ginger, our assessment only included those clinical trials that had been conducted since the publication of these SRs. We also analyzed the findings of these SRs [29,30,31] (Table 4).

In their SR, Nikkhah Bodagh et al. concluded that a daily dosage of 1500 mg of ginger is effective in relieving pregnancy induced nausea and vomiting (PINV), and that more investigation is needed before making a similar recommendation for other GI conditions due to the paucity of evidence [56]. Upon closer examination, however, this SR did not report any research methodology regarding how they conducted the SR; among other details, information indicating how they arrived at the dosage recommendation is lacking. Additionally, the included data set is heterogeneous, utilizing findings from RCTs, SRs, and meta-analyses (MAs), and considering the various etiologies of nausea simultaneously, integrating heterogenous outcomes and thus weakening the quality of its evidence.

Anh et al.’s SR reviewed 109 RCTs focused on clinical use of ginger, 47 of which were related to ginger’s antiemetic function and three to its gastric emptying function [57]. While we did not examine each included reference in depth due to the data-set size, we concluded that their system for rating quality of evidence was sufficient based on their use of the Cochrane Collaboration tool and granularity in research methodology reporting. Regarding their assessment of ginger’s gastric emptying ability, the three RCTs reported inconsistent effects [58,59,60]. Additionally, only one of these studies was rated by the authors as “high quality evidence” and upon closer examination, this study involved a crossover design. Regarding their assessment of articles analyzing various types of nausea, they concluded that a divided daily dosage of less than or equal to 1500 mg ginger could be recommended for PINV due to all 10 reported RCTs yielding consistent data, but that more trials are needed to ascertain results for chemotherapy induced nausea and vomiting (CINV) and post-operative nausea and vomiting (PONV) due to conflicting results.

Three trials examining ginger’s effectiveness were published since these SRs: Attari et al. and Panda et al. examined FD; Bhargava et al. examined upper GI symptoms in advanced cancer patients (Table 5) [29,30,31]. While Attari et al. and Panda et al. examined FD, these studies utilized inconsistent daily dosing (3 g versus 400 mg), extract composition, use of ginger powder versus OLNP-06 (standardized high gingerol concentration), and heterogenous subjective reporting questionnaires, making it difficult to integrate outcomes and ascertain an effective dose. Additionally, Attari et al.’s study examined *H. pylori* positive FD subjects; although *H. pylori* is also a necessary area for study due to its high global prevalence and potential role as a factor in the development of dyspepsia, it is challenging to elucidate the impact of ginger on FD alone based on this study, and is therefore difficult to integrate its outcomes [61,62]. Furthermore, this study did not include a placebo and had a small sample size (n = 15). Conversely, Panda et al. conducted a higher powered trial (n = 50) with proper blinding and a placebo. Panda et al. reported two mild AEs in the intervention group and Attari et al. did not provide AE information. Based on this, ginger shows promise in managing FD symptoms such as heartburn, nausea, and early fullness, but requires larger, appropriately designed clinical trials with standardized interventions to reach consensus on a consistent ginger extract composition and dosage.

Bhargava et al. found that ginger (1650 mg per day) significantly improved upper GI symptoms such as dysmotility-, reflux-, and ulcer-like symptoms, as well as nausea and anorexia in advanced cancer patients (n = 14) with anorexia-cachexia syndrome [31]. In addition, nine of the fifteen patients saw improvement in stomach gastric myoelectrical activity, indicated by an electrogastrogram (EGG), which could be related to motility. The authors noted their study limitations, such as the small sample size, short duration (14 days), and lack of a placebo. In sum, this study indicates the need for further trials to confirm ginger’s application for dyspepsia-like symptoms in advanced cancer patients. Whether these results can be generalized to non-cancer patients also needs further investigation.

Collectively, it appears that a divided daily dosage of 1500 mg of ginger may be effective in reducing PINV but that further trials are needed to reach a consensus on a dosage and standardized extract composition for other types of nausea besides FD [63].

##### Licorice

There is a paucity of trials examining the MoAs underlying licorice’s effectiveness for FD. Animal trials attribute its anti-ulcerogenic activity to observed free radical scavenging, as well as mucus production, and prostaglandin inhibition [32,64,65,66]. Licoflavone and glycyrrhizin are commonly indicated as the active ingredients responsible for the observed effects [67,68].

Two clinical studies were identified that investigated the effects of licorice on upper GI symptoms (Table 5) [32,33]. Prajapati and Patel reported that licorice root powder and licorice substitute (*T. nummularia*) (both dosed at 2 g, three times per day) were effective in significantly reducing indigestion symptoms in subjects (n = 40) such as eructation with a bitter or sour taste, nausea, and a burning sensation in the throat and chest [32]. In addition to lacking a placebo, this study did not state whether it was blinded, and seemed primarily interested in investigating the substitute’s effectiveness.

In a double-blind RCT, Raveendra et al. observed a significant improvement in dyspeptic symptoms and QoL in Rome III classified FD subjects (n = 50) receiving a standardized flavonoid-rich licorice extract (75 mg, twice per day) over 30 days as compared to placebo [33]. Although they utilized a validated questionnaire measuring QoL, their method for assessing improvements in FD symptoms was a subjective measure. No AEs were reported.

##### Papaya

Animal and in vitro studies suggest that papaya has an ability to scavenge ROS and inhibit gastric secretion through histamine reduction [69,70,71] and that it also acts directly upon gastric smooth muscle to impact its motility [72].

Muss et al. and Weiser et al. examined papaya extracts among subjects with a range of GI dysfunctions such as constipation, heartburn, and symptoms of irritable bowel syndrome (IBS) and chronic gastritis (Table 5) [34,35]. Muss et al. found that a papaya proprietary blend (20 mL per day; standardized to higher papain activity) improved heartburn symptoms non-significantly (n = 13) [34]. Weiser et al. observed an improvement in upper GI complaints in patients (n = 60) supplemented with papaya formula (20 g, twice per day; formula containing papaya and oats) as compared to a placebo, but these differences were non-significant [35]. Muss et al. did not reveal AE data and Weiser at al. reported zero AEs. Although both studies examined papaya, they used widely different formulations and are thus difficult to compare.

#### 3.2.3. Combination Products

Two studies explored the relationship between combinations of various botanical ingredients and their impact on upper GI conditions (Table 6) [44,73]. In addition, we also discuss the recent SR summarizing the effectiveness of peppermint oil and caraway oil (POCO).

Ried et al. studied the effects of curcumin, A. vera, slippery elm, guar gum, pectin, peppermint oil, and glutamine in three amounts (5 g/day; 10 g/day; choice of 0–10 g/day) for three consecutive, four-week trial periods on adults (n = 43) experiencing one or more upper and/or lower GI symptom at least once per week for at least 3 months. The authors found a significant improvement in indigestion, heartburn, regurgitation, nausea, and QoL, as well as a decrease in GERD symptom frequency and incidence (Table 6) [73]. The authors reported two mild AEs related to the trial (severe bloating and constipation) which resulted in dropout. While the study utilized a 4-week run-in period to act as the control, the intervention period was a single-arm design and lacked a placebo. Slippery elm has received increased interest, but trials examining its singular effect are lacking, with Ahuja and Ahuja highlighting it as a promising ingredient due to its raft-forming potential and therefore its ability to ameliorate GERD symptoms, but stating that this requires further investigation [74].

Brown et al. conducted a crossover RCT trial in which participants (n = 24) consumed a specific high-fat, reflux inducing meal followed by 30 min of chewing one of the two treatment gums, either the placebo gum or the intervention gum, containing calcium carbonate (500 mg), ACV, licorice, and papain (Table 6) [44]. The intervention significantly reduced heartburn and mean acid reflux score (measured by a symptom based visual analogue scale (VAS)) in GERD patients (n = 24) as compared to the placebo. Given the combinatorial nature of the intervention, as well as calcium carbonate’s effects on GERD symptom reduction and the known heartburn-reducing effect of chewing gum alone, it is challenging to elucidate each individual ingredient’s role in the observed outcome [75].

##### Peppermint Oil and Caraway Oil

Peppermint oil (PO) has been a long-studied ingredient for its possible effects on upper GI conditions as well as the combined formulation of POCO. A recent paper reviewed peppermint’s effects on FGIDs, finding that PO was always used in combination with additional ingredients when used for FD, either as POCO or another formulation [76]. The three RCTs they included consisted of heterogenous combinations and outcomes, with two showing improvement in FD symptoms and the other showing no significant difference between the intervention and a placebo. Another recent SR-MA also examined POCO’s effectiveness in FD, including five RCTs (n = 578) which were of the same duration (4 weeks) examining heterogenous formulations [77]. The authors’ use of the Cochrane Collaboration risk of bias tool was satisfactory. They found that POCO resulted in statistically significant global improvement in FD symptoms and improvement in epigastric pain. The common divided daily dosage was one capsule of 90 mg PO and 50 mg CO, twice daily. This SR-MA concluded by calling for higher quality studies investigating longer term effects of POCO on FD. In addition, there is a need for trials studying the effects of isolated PO on FD. The underlying MoA for POCO’s effects on FD is unclear, but may have to do with POCO’s observed ability to modulate post-inflammatory visceral hypersensitivity in rats [78] as well as its effect on decreasing frequency and amplitude of the migrating motor complex (MMC) in healthy volunteers [79].

### 3.3. Non-Botanical Ingredients Addressing Heartburn, GERD, and Gastric Conditions

Two studies investigated the impact of non-botanical nutritional ingredients: activated charcoal (AC) and zinc-l-carnosine (ZnC) on upper GI conditions (Table 7) [80,81]. In addition, we mention the recent work dedicated to melatonin.

#### 3.3.1. Activated Charcoal

Lecuyer et al. (n = 132) and Coffin et al. (n = 276) found that two formulations of AC and simethicone with or without magnesium oxide resulted in a significant reduction in FD symptoms in FD patients as compared to placebo (Table 7) [80,81]. Coffin et al. reported four patient dropouts in both groups because of mild AEs related to GI symptoms while Lecuyer et al. reported zero AEs.

Most studies on AC examine its relationship to lower GI gas. Furthermore, most of these trials were conducted in the 1990s, are relatively low powered, and yielded conflicting results, with some indicating that AC significantly reduced breath hydrogen levels and GI symptoms, and others not showing any effect [83,84,85,86]. These studies all indicated unclear mechanisms of action, though authors hypothesized that AC likely either adsorbed the gas or gas-causing agents or prevented gas formation. Although this benefit might seem relevant only to lower GI symptoms, Lecuyer et al. make the point that dyspeptic symptoms such as early satiety and bloating might also be related to the accumulation of air. Thus, one could hypothesize that AC could reduce the dyspeptic symptoms by mitigating abdominal air [80]. However, the formulations studied could also both be effective simply due to the use of simethicone [80,87,88]. As a result, more extensive research needs to be performed to assess the efficacy of AC alone.

#### 3.3.2. Melatonin

While the use of melatonin as a sleep aid has been well established, research also indicates its potential role in GERD and FD [89,90]. While an SR has not yet been conducted, Bang et al. composed a protocol for an SR-MA assessing melatonin’s effectiveness in the context of GERD [91].

A trial studied patients (n = 36) with and without GERD, who received the following four interventions: a control; melatonin (3 mg, once per day at bedtime); omeprazole (20 mg, twice per day); and a combination of omeprazole and melatonin. They found that the two groups receiving melatonin supplements alone, or in addition to omeprazole, yielded a significant increase in LES pressure, while the groups receiving omeprazole alone or in addition to melatonin yielded a significant increase in pH and serum gastrin levels, and a significant decrease in basal acid output (BAO), compared to melatonin alone [90]. While they utilized diagnostic tools such as esophageal manometry, pH-metry, BAO monitoring, and serum measurements to assess outcomes, the authors did not share their methods for assessing improvements in heartburn and epigastric pain. An RCT by Klupinska et al. examined the role of melatonin (5 mg, once per day at bedtime) versus placebo in FD subjects (n = 60) and found that 56.6% of the melatonin group had complete resolution of dyspeptic symptoms and 30% experienced a partial resolution, while the majority of placebo subjects (93.3%) saw no improvement [89]. Additionally, those in the melatonin group used significantly less alkaline relief tablets than those receiving the placebo. The authors utilized a Likert scale to rate pain, sleep, and symptom intensity. MoA have been proposed for melatonin’s effectiveness in upper GI conditions, including animal trials that have demonstrated decreased secretion of HCl and pepsin [92], and also the stimulation of gastrin release, which could mitigate reflux by increasing the contractile activity of the LES [93].

#### 3.3.3. Zinc-l-carnosine

ZnC, also known as polaprezinc, a chelate of zinc with L-carnosine, is a compound with numerous clinical trials conducted in the 1990s, and a long history of use in Japanese medicine as a gastric mucosal protective agent [82,94,95,96,97]. In an RCT, Tan et al. found that all three arms consisting of ZnC (either 75 mg or 150 mg, 2x per day), in combination with triple therapy (omeprazole (20 mg), amoxicillin (1 g), clarithromycin (500 mg), each twice daily) or triple therapy alone, were effective in significantly reducing symptoms in *H. pylori* positive gastritis patients, including abdominal pain, acid reflux, belching, heartburn, bloating, nausea, and vomiting (Table 7) [82]. As mentioned above, as *H. pylori* infection has a high global prevalence and can be an underlying risk factor for the development of dyspepsia symptoms, it is an important area to also investigate [61,62]. However, as there are no non-H. pylori gastritis studies examining ZnC, it is not clear what the results could be regarding ZnC’s effectiveness for heartburn or dyspepsia. Additionally, as this study did not include a placebo, it is difficult to elucidate ZnC’s superiority over prescriptions used for gastritis. While MoAs related to gastritis have not been elucidated, animal and in vitro studies have attributed ZnC’s observed anti-ulcerogenic activity to binding and healing of the ulcer site, as well as its ability to reduce levels of inflammatory cytokines, increase heat shock proteins, and inhibit gastric TNF-α concentration [95,98,99,100,101].

### 3.4. Areas for Improvement of Future Clinical Trials

Due to the many challenges involved in designing rigorous trials utilizing nutritional supplement ingredients, the studies included in this review shared many common issues, which should be addressed during future research. We will highlight those areas here in the hopes of guiding future clinical trials in the service of more accurate and reproducible results.

As many trials lacked proper controls, utilizing prescription or OTC medications, designing future trials that include the use of a placebo is paramount. Additionally, trials enlisting a larger sample size and longer duration are needed, as most trials included in this review had less than 100 subjects and a duration of less than 30 days. Regarding the intervention, determination of a standardized extract composition and dosage is needed, as nutritional compounds often contain many component ingredients. Identifying the active ingredient(s) therein and their associated MoAs is essential for the reproducibility of clinical study results. Furthermore, many of these trials examined ingredients in combination; it is necessary to develop trials examining ingredients in isolation in order to elucidate the specific effect of each individual ingredient.

Controlling for diet is another important feature of rigorous study design; most studies analyzed herein took a heterogenous approach, with some making no mention of diet implementation and others totally changing subjects’ diets to one that would either increase or decrease reflux incidents. It could also be important to assess whether subjects’ dietary intake already involves the bioactive agent of interest, e.g., study subjects who are already on high-fiber diets.

Additionally, a standardized protocol around the usage of prescription and OTC medications before and during the trial would help ensure that results could be integrated between studies and avoid confounding variables. Ensuring that studies are double-blinded and utilize allocation concealment is likewise paramount to optimize the quality of the evidence they can provide. To ensure consistency and reproducibility, studies would do well to use specific validated questionnaires for interval symptom assessments and to use standardized diagnostic criteria for the purposes of recruitment.

## 4. Conclusions

This review highlights both those nutritional ingredients with robust evidence for managing upper GI conditions and those that, although traditionally associated with upper GI conditions, lack significant data at present. Those with robust evidence include ginger for pregnancy induced nausea and vomiting, and peppermint and caraway oil for functional dyspepsia. Melatonin and marine alginate appear promising for GERD use; a comprehensive systematic review assessing the current body of evidence for melatonin and additional trials investigating marine alginate against a placebo would be helpful to determine their effectiveness and the appropriate dose for each. Fenugreek, galactomannan, and zinc-l-carnosine represent promising areas for further study. Additionally, A. vera, papaya, and licorice are each represented in a handful of trials, but mostly exist in a combinatorial form; further investigation of each ingredient in isolation would help to elucidate their specific effects. The remainder of the nutritional ingredients examined in this review lack substantial data to support their use for upper GI symptoms. While attending to those ingredients with relatively stronger evidence for current clinical applicability, we believe this review may help set a focused research agenda for additional studies moving forward, calling for trials examining dose–response using standardized extract formulations and attending to specific symptomatic benefits.

## Figures and Tables

**Figure 1 nutrients-14-00672-f001:**
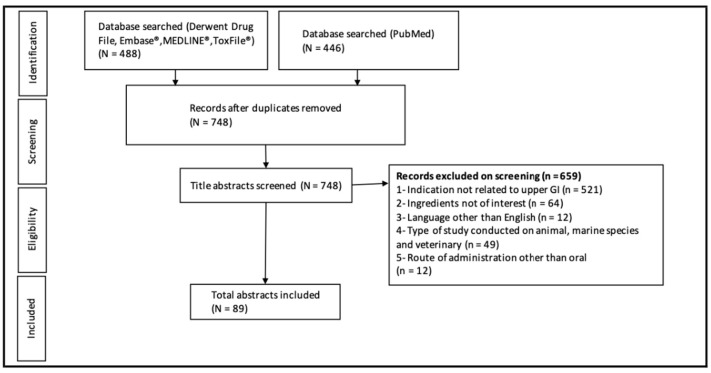
Flow diagram of the literature search process.

**Table 1 nutrients-14-00672-t001:** Ingredients of interest with alternative naming schemes.

Ingredient	Alternative Name
Activated Charcoal	Activated Carbon
Aloe Vera	*Aloe barbadensis* Miller; *Aloe aborescens*
Apple Cider Vinegar	ACV
Ashwagandha	*Withania somnifera*
Burdock Root	*Arctium lappa*
Cardamom	*Zingiberaceae e* *lettaria; Zingiberaceae amomum*
Chamomile	*Matricaria recutita*
Chicory	*Cichorium intybus*
Clove Oil	*Syzygium aromaticum*
Dandelion	Taraxacum officinale
D-Limonene	Limonene
Fennel	*Foeniculum vulgare*
Fenugreek	*Trigonella foenum-graecum*
Galactomannan	*-*
Ginger	*Zingiber officinale* Roscoe
Gum Arabic	Acacia gum
Lemon Balm	*Melissa officinalis*
Licorice	*Glycyrrhiza glabra*
Marine Alginate	Alginate
Melatonin	N-acetyl-5-methoxytryptamine
Papaya	*Carica papaya*, papain, papaw
Partially Hydrolyzed Guar Gum	PHGG
Peppermint	*Mentha pipereta*
Slippery Elm	*Ulmus rubra*
Zinc-L-Carnosine	Z-103, Polaprezinc, L-CAZ, N(3 aminopropionyl)-L-histidine

**Table 2 nutrients-14-00672-t002:** PICOS criteria for inclusion and exclusion of studies.

Parameter	Criteria	Exclusion
Population	Healthy and occasionally or chronically ill adults (≥18 years)	Individuals <18 years, animals, in vitro
Intervention	Galactomannan, Gum Arabic, Partially Hydrolyzed Guar Gum, Fenugreek, Zinc Carnosine, Chicory, Burdock Root Arctium, Slippery Elm, Activated Charcoal, Clove Oil, Papaya, Cardamom, Ginger, Fennel, Aloe, Chamomile, Lemon Balm, Dandelion, Ashwagandha, Peppermint, Marine Alginate, Melatonin, Apple Cider Vinegar, Licorice, and D-Limonene	N/A
Comparator	Placebo, control, none, or standard care	N/A
Outcome	Improvement of upper GI symptoms	N/A
Study Design	RCTs, clinical studies, and review articles	Case reports, observational studies, editorials, comments, notes, and letters

**Table 3 nutrients-14-00672-t003:** Search string for nutritional ingredients and upper GI conditions and symptoms.

S. No	Searched for	Databases
1	Key words used for intervention: ti,ab(“Withania somnifera” OR Ashwagandha OR Taraxacum OR “yellow flowers” OR Dandelion OR “Melissa officinalis” OR “Lemon balm” OR Camomile OR Chamomile OR “Matricaria chamomilla” OR Aloe OR “Foeniculum vulgare” OR fennel OR “Zingiber officinale” OR Ginger OR cardamon OR cardamum OR Elettaria OR “Carica papaya” OR papaw OR papain OR “papaya” OR eugenol OR “clove oil” OR “Activated charcoal” OR “Ulmus rubra” OR “Slippery elm” OR “Burdock root “ OR Arctium OR “Cichorium intybus” OR “Zinc carnosine” OR “Trigonella foenum” OR “gum sudani” OR “acacia gum” OR “Arabic gum” OR “gum acacia” OR “Senegal gum” OR “Gum Arabic” OR “galactomannan” OR “Partially hydrolyzed guar gum” OR “Mentha piperita” OR “peppermint oil” OR “marine alginate” OR “alginate” OR “Melatonin” OR “N-acetyl-5-methoxytryptamine” OR “D-limonene” OR “Limonene” OR “Apple cider vinegar” OR “ACV” OR “Licorice” OR “Glycyrrhiza glabra” OR “Liquorice” OR “DGL” OR “deglycerized licorice” OR “deglycyrrhizinated licorice” OR “deglycyrrhizinated liquorice” OR “Liquiritiae radix”)	Embase, Medline, Derwent drug file, PubMed and ToXfile
2	Indications ti,ab(“gastric gas” OR belching OR Eructation OR Dyspepsia OR indigestion OR “gastroesophageal reflux” OR pyrosis OR Heartburn OR satiety OR overfeed OR Overfill OR “difficulty in swallowing” OR “abdominal bloating” OR “appetite loss” OR distention OR Nausea OR vomit* OR burp* OR dysphagia OR “stomach spasm” OR “stomach pain” OR “Stomach fullness” OR flatulence OR “stomach gas” OR “upper gastrointestinal” OR bloating OR “postprandial fullness” OR “loss of appetite” OR “food regurgitation” OR acidity OR retching OR gagging OR “motion sickness” OR GERD)	Embase, Medline, Derwent drug file, PubMed and ToXfile

**Table 4 nutrients-14-00672-t004:** Summary of human trials on botanical fiber ingredients and upper GI conditions.

Nutritional Ingredient	Reference	Population (n)	Characteristics	Study Design	Duration	Intervention	Control/Comparator	Outcome Measured	Results
Fenugreek	DiSilvestro et al., 2011 [14]	n = 45	Age 43 ± 8; 24 females and 21 males; subjects experiencing heartburn after 3–8 meals per week for at least a month	RCT; placebo CG and IG were blinded, ranitidine positive CG group was unblinded; rescue medication tablets of chewable calcium carbonate were allowed	2 weeks	IG: 2000 mg 2x/day fenugreek capsules (4 g daily dose)	Placebo CG: starch capsules, four capsules taken twice/day; Positive CG: Ranitidine (Zantac 75), 75 mg, 2x/day, (150 mg daily dose)	Heartburn	The severity and incidence of heartburn significantly decreased with both the IG and the positive CG for both the first and second intervention week. The placebo also yielded significant effects for the second but not the first intervention week. All three interventions yielded statistically significant reductions in rescue medication use.
Galactomannan	Abenevoli et al., 2021 [15]	n = 60	Age ≥18; 47 males and 13 females; adults with GERD symptoms not taking PPIs	Single-center, RCT	2 weeks	IG: 10 mL, 3x/day liquid blend (calcium carbonate, sodium bicarbonate, *Malva sylvestris,* hyaluronic acid) (30 mL total)	CG: Placebo, one sachet containing 10 mL liquid, three times per day (30 mL total)	GERD	100% of patients reported at least 30% reduction in symptoms from baseline to week 3 of the trial (*p* < 0.001) as compared to placebo group. Symptom frequency and intensity were progressively significantly reduced from baseline to visit 4 (*p* < 0.001) in the IG as compared to placebo. Heartburn significantly decreased from baseline to day 14 for the IG (*p* < 0.001) versus an increase in score in CG (*p* > 0.05). There was a significant decrease in GERD symptoms from visit 2 to 3 to 4 (*p* < 0.001, *p* < 0.001, and *p* = 0.001 respectively) as compared to the CG.

IG = intervention group, CG = control group, RCT = randomized control trial.

**Table 5 nutrients-14-00672-t005:** Summary of human trials on “other” botanical ingredients and upper GI conditions.

Nutritional Ingredient	Reference	Population (n)	Characteristics	Study Design	Duration	Intervention	Control/Comparator	Outcome Measured	Results
Aloe Vera	Panahi et al., 2015 [28]	n = 79	Age: 18–65; 45 females and 34 males; GERD patients	RCT	4 weeks	IG: 10 mL 1x/day A. vera syrup (standardized to 5.0 mg polysaccharide per mL of syrup) (10 mL total)	CG1: omeprazole capsule (20 g once a day) CG2: ranitidine tablet (150 mg in a fasted state in the morning and 150 mg 30 min before sleep at night) (300 mg total)	GERD	A. vera was effective in significantly reducing the frequencies of all GERD symptoms except vomiting at weeks 2 and 4 for within-group comparison to baseline (*p* < 0.05). Omeprazole and ranitidine were both more effective at reducing heartburn and flatulence than A. vera. The frequency of other symptoms did not differ between the groups.
Aloe Vera	Panahi et al., 2016 [27]	n = 85	Age >40 years; male GERD veterans with sulphur mustard gas exposure	RCT	6 weeks	IG: pantoprazole (40 mg before breakfast) plus 5 mL, 2x/day A. vera syrup (10 mL total)	CG: pantoprazole (40 mg before breakfast)	GERD	Both IG and CG resulted in a progressive decrease in RSI score from baseline to weeks 3 and 6 ((*p* < 0.001) with the IG having a significantly greater reduction (*p* < 0.001) compared to CG. No adverse events were reported.
Ginger	Panda et al., 2020 [30]	n = 48	Age: 18–55; subjects with FD per Rome III criteria	RCT, parallel group	4 weeks	IG: 200 mg, 2x/day high concentration gingerol powder extract (400 mg total)	CG: placebo, 200 mg twice daily	FD symptoms	The IG had significantly more subjects who were “extremely” or “markedly” improved as compared to CG. (79% vs. 21%; *p* < 0.05). Elimination rate of symptoms both individually and collectively was greater in the IG than CG (64% vs. 13% of subjects) (*p* < 0.05).
Ginger	Attari et al., 2019 [29]	n = 15	Age: 18–65; 5 males and 10 females; patients with *H. pylori* positive FD	Pilot study	4 weeks	IG: 3 g, 1x/day ginger powder tablets (3 g total)	CG: none	*H. pylori* positive FD and FD	Ginger supplementation resulted in significant improvement of all dyspepsia symptoms including fullness, early satiety, nausea, belching, gastric pain, and gastric burn, but not vomiting (*p* = 0.180).
Ginger	Bhargava et al., 2020 [31]	n = 15	Age: 35–79; 8 males and 7 females; patients with anorexia-cachexia syndrome (ACS) in addition to a variety of advanced cancer diagnoses	Single-arm intervention trial	2 weeks	IG: 1650 mg 1×/day of ginger powder capsule (1650 mg total)	CG: none	ACS GI symptoms such as nausea, vomiting, dysmotility-, ulcer-, and reflux-like symptoms	Over half of the patients reported significant improvements in GI symptoms including nausea (*p* < 0.02), dysmotility-like (*p* < 0.01), reflux-like (*p* < 0.01), and ulcer-like symptoms (*p* = 0.05).
Licorice	Prajapati and Patel, 2015 [32]	n = 40	Age: 21–60; Amlapitta (acid gastritis) patients including symptoms of indigestion, exhaustion, eructation with bitter or sour taste, burning sensation in the chest and throat, and anorexia	RCT	2 weeks	IG1: 2 g, 3×/day of Licorice root powder (6 total) IG2: 2 g, 3×/day of *Jethimala* (Taverniera nummularia Baker) (commercial licorice substitute)	CG: none	Gastritis; heartburn; anorexia; reflux	Licorice root and *Jethimala* were both effective in significantly reducing all symptoms in both groups, with licorice treatment showing overall better effects. There was no significant difference between the two treatments in treatment efficacy (*p* > 0.05) other than in the symptom of anorexia (*p*-0.001).
Licorice	Raveendra et al., 2012 [33]	n = 50	Age: 18–65; 31 males and 19 females; patients with FD as diagnosed by Rome III criteria	RCT	30 days	IG: 75 mg, 2×/day of flavonoid-rich extract of licorice (150 mg total)	CG: placebo	Functional dyspepsia	As compared to CG, IG showed a significant decrease in total symptom scores (*p* ≤ 0.05), and a significant improvement in quality of life (*p* ≤ 0.05) as well as overall treatment efficacy.
Papaya	Muss et al., 2013 [34]	n = 84	Age: 18–75; subjects with dysfunctions of the GI tract such as constipation, heartburn, and irritable bowel syndrome (IBS)	RCT; participants were labeled as “early” (2 days) or “late” (3–16 days) returnees based on when they returned to the trial center to complete the endpoint questionnaire	40 days	IG: 20 mL, 1×/day papaya formulation (standardized to higher papain activity)	CG: 20 mL, 1×/day of placebo	Heartburn, constipation, and bloating	In the “early returnees,” the IG showed significant improvements in symptoms of constipation (*p* < 0.031) and flatulence (*p* = 0.017) as compared to placebo. Regarding heartburn, 85% of evaluable participants (n = 13) reported improvement (*p* = 0.114). These effects vanished in the “late returnees,” but those in the IG showed more beneficial effects than the CG.
Papaya	Weiser et al., 2018 [35]	n = 60	Age: 18–75; 22 males and 38 females; patients with endoscopically confirmed chronic gastritis	RCT	30 days	IG: 20 g, 2×/day of papaya blend before meal (papaya pulp, organic whole meal oat flour, apple juice concentrate, natural aroma, and water)	CG: 20 g, 2×/day placebo before a main meal	Chronic Gastritis	There was a reduction in all symptoms in both the IG and CG with greater reduction in scores for the IG for acute stomach ache pain, pain severity, impact on daily routine, nausea, bloating, and pain in the upper abdomen, but no significant difference between the groups. The only symptom which was significantly reduced in the IG compared to CG was pain load (*p* = 0.048).

ACS: Anorexia Cachexia Syndrome.

**Table 6 nutrients-14-00672-t006:** Summary of human trials on botanical combination ingredients and upper GI conditions.

Nutritional Ingredient	Reference	Population (n)	Characteristics	Study Design	Duration	Intervention	Control/Comparator	Outcome Measured	Results
Curcumin, aloe vera, slippery elm, guar gum, pectin, peppermint oil, and glutamine	Ried et al., 2020 [73]	n = 43	Mean age: 50; 76% of participants were female; adults with moderate upper and/or lower GI disturbances	Single-arm pre-post study	16 weeks (4-week run-in period and 12-week intervention period)	IG: 5 g, 1×/day formula mixed with water for 4 weeks, followed by 10 g/d for the second month and finally the patient’s preferred dose (0/5/10 g/d) for the third month (Curcumin, Aloe vera, slippery elm, guar gum, pectin, peppermint oil, and glutamine)	CG: 4-week run-in period	Upper and lower GI symptoms	There was a significant improvement of upper GI symptoms including indigestion, heartburn, regurgitation (acid reflux), and nausea (*p* < 0.001), overall decreased upper GI pain (*p* < 0.001) as well as improved quality of life (QoL) (*p* < 0.001) after 12 weeks.
ACV, Licorice, papain	Brown, et al., 2015 [44]	n = 24	Age ≥18; mean age: 34 ± 14; 17 females and 7 males; GERD patients	Double-blind, placebo controlled, crossover trial with 1-week washout between treatments; reflux causing meal: big hamburger, French fries, hot sauce, soda	Single day intervention	IG: 30 min chewing intervention gum following reflux causing meal (gum active ingredients: calcium carbonate (500 mg), licorice extract, papain, and apple cider vinegar)	CG: 30 min chewing placebo gum following reflux causing meal	GERD and heartburn	Adjusted mean heartburn score and mean acid reflux score were significantly decreased in IG as compared to CG (*p* = 0.034 and *p* = 0.013 respectively). There were no significant differences between groups for pain, nausea, and belching, although they trended towards greater improvement in the IG group.

ACV = apple cider vinegar.

**Table 7 nutrients-14-00672-t007:** Summary of human trials on non-botanical ingredients and upper GI conditions.

Nutritional Ingredient	Reference	Population (n)	Characteristics	Study Design	Duration	Intervention	Control/Comparator	Outcome Measured	Results
Activated Charcoal (AC)	Coffin et al., 2011 [80]	n = 276	Age: 18–49; mean age 39 ± 10 years; 70% female; functional dyspeptic patients per ROME III criteria	RCT, phase III trial	30 days	IG: 2 capsules 3x per day of AC formula (gastro-soluble capsule containing 140mg of AC, 45mg of simethicone INN, and 180mg of magnesium oxide, plus enteric coated capsule containing 140mg of AC and 45mg of simethicone INN)	CG: 2 placebo capsules, 3x per day	Functional dyspepsia	IG saw significantly greater absolute and relative symptom reductions compared to placebo. The IG observed a significant reduction in post-prandial fullness (*p* = 0.034), epigastric pain (*p* = 0.045), epigastric burning (*p* = 0.03), and bloating (*p* = 0.03), in comparison to placebo, and early satiety approached significance (*p* = 0.051).
Activated Charcoal	Lecuyer et al., 2009 [81]	n = 132	Age: 18–49; mean age: 39.0±8.8 years; functional dyspepsia patients	RCT	3 months and 2-month follow-up period	IG: 2 capsules, 3x per day AC formula during meals (gastro-soluble and enteric capsules containing 140 mg AC and 45 mg simethicone)	CG: 2 capsules placebo, 3x per day during meals	Functional dyspepsia	Greater percentage of patients with a reduction of at least two points on the symptoms intensity scale in IG compared to CG (*p* = 0.043) although there was a nonsignificant difference between the two groups regarding overall patient complaints (*p* = 0.115). IG also saw greater reduction in intensity of abdominal fullness, bloating, and the sensation of slow digestion versus placebo (*p* < 0.05).
Zinc-l-Carnosine (polaprezinc) (ZnC)	Tan et al., 2017 [82]	n = 303	Age: 18–70; 168 females and 164 males; patients with *H. pylori*-associated gastritis	RCT	2 weeks	IG: Arm A: triple therapy (omeprazole 20 mg, amoxicillin 1 g, and clarithromycin 500 mg, each twice daily) plus polaprezinc 75 mg, twice daily (150 mg total); Arm B: triple therapy plus polaprezinc 150 mg, twice daily (300 mg total)	CG: Arm C triple therapy alone	*H. pylori* associated gastritis	All three arms saw significant gastrointestinal symptom improvement, including abdominal pain, acid reflux, belching, heartburn, bloating, nausea, and vomiting at days 7, 14, and 28 when compared to baseline (*p* < 0.0001) with no significant difference between groups. Both the intention to treat (ITT) and per-protocol (PP) analyses showed that Arms A and B had a significantly higher rate of *H. pylori* eradication than Arm C while there was no significant difference between the rate of eradication in Arms A and B.

AC: Activated charcoal, ZnC: zinc-l-carnosine.

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
