# Peer review of "Effectiveness of Nutritional Ingredients on Upper Gastrointestinal Conditions and Symptoms: A Narrative Review"

_nutrients, 2022, doi:10.3390/nu14030672_

Round 1
Reviewer 1 Report
I read this article with great interest. I believe this is a well done review of the currently literature.
- The study on fenugreek in the text reads a bit confusing as it almost seems fenugreek and ranitidine were used simultaneously. Could you please try and make it a little more clear that fenugreek was compared to placebo and to ranitidine.
Otherwise I think this is a nice review.
Author Response
Thank you for your comments and insight.
We have changed the fenugreek paragraph to reflect your recommendation so as to ensure the reader understands that fenugreek and ranitidine groups were separate. Please see the below sentence marking this distinction.
"DiSilvestro found that the fenugreek fiber group (2000 mg, twice per day, standardized to contain 85% galactomannan), and the ranitidine group (75 mg, twice per day) both yielded reduced heartburn severity and incidence in subjects (n=45) over two weeks and that fenugreek was significantly more effective than placebo."
.
Reviewer 2 Report
The present review manuscript is well written and provides valuable information about the nutritional ingredients on upper gastrointestinal conditions and symptoms.
This review article discusses in details some ingredients of interest and their use for upper GI support. Whatever, the list could have been extended, e.g. related to gums, only gum arabic, and marine alginate were discussed, but on the top list of interest are guar gum, xanthan gum, carrageenan.
Based on which reason only the 25 nutritional ingredients have been chose and why only them? where is the novelty of? Line 15-17 do not support the reason
The conclusion is adequate, and the used references are sufficient and up-to-date.
Author Response
Thank you for your recommendations and insight on our review.
In response to the reasoning behind why we chose the specific 25 ingredients, it had to do with what the literature was currently pointing to in terms of effectiveness in the upper GI space. We looked into the gums that you mentioned including guar gum, xanthan gum, and carrageenan, but were unable to find sufficient material to expand upon/summarize in the upper GI space. Lines 185-186 point to this finding and we agree with you that this will be an excellent area for future study. We appreciate your suggestion but due to the above findings as well as the need to limit our scope, we arrived at the 25 ingredients, and weren't able to use novelty as a discriminator.
We value your input and review on this piece.
Reviewer 3 Report
The article does not comply with the requirements for writing the journal (the abstract has 330 words, the references are at the limit, the formatting of the tables is not timely, the data in the tables 4,5,6,7,8 is difficult to track, the text is not misaligned justified, the citation is incorrect). These minimum non-compliant requirements provide clues as to the rigor of the approach.
Line 104 –I suggest moving Table 3 in Supplementary Materials.
Line 145; line 412; line 481 different marking.
A summary table is missing for the sources mentioned on Botanical/ Non-Botanical ingredients addressing gastric conditions.
The review results reported are too premature for publication. More work is needed to substantiate the conclusions in this manuscript.
Author Response
Thank you for your feedback on our review.
We have changed the abstract to fit the 300 word limit parameter but the other manuscript formatting guidelines have been followed. We included a description on our submission on why the tables are not timely interspersed due to table size and this was understood by our assigned editor.
We appreciate your thoughts on moving table 3 to supplementary material but have decided to keep it in the text at this time.
We see your comment regarding Line 145; line 412; and line 481 and these lines all mark the different sections as do the other headings.
The summary table addressing botanical/ non-Botanical ingredients addressing gastric conditions includes table 5-8.
Thank you for your comment highlighting that some of the data is new in its substantiation. We appreciate you highlighting this. However, there are few studies published in this area and thus, there is a need to summarize the results.
Round 2
Reviewer 3 Report
I would like to thank the authors for addressing my initial comments.